# Social Model—Innovation and Behavioural Intervention as a Public Policy of Action within an Oncology and Loneliness Scope

**Vasco Fonseca** [1,2,*] [iD], **Joaquim Caeiro** [1] [iD] **and Fernanda Nogueira** [1]

1   Centre for Public Administration and Public Politics, Institute of Social and Political Sciences, University of Lisbon, 1300-663 Lisbon, Portugal; jcaeiro@iscsp.ulisboa.pt (J.C.); fnogueira@iscsp.ulisboa.pt (F.N.)
2   Centro Hospitalar de Lisboa Ocidental, E.P.E.—Hospital de São Francisco Xavier, 1449-005 Lisbon, Portugal
*   Correspondence: medicinavf@yahoo.com

**Abstract:** The article presents concepts and the Public Health Policy University of Lisbon Lab project to answer questions about the macro-environment of cancer and loneliness. Although the biomedical model has considered the disease's general symptoms, it takes a holistic approach to incorporate several other circumstances that influence health. Emotional, social, psychological, and economic factors mirror influencing layers that affect wellness. Portugal follows Europe's tendency and simultaneously reflects its reality. Governmental internal policies, amplified by regulations, improve disease prevention and treatment. Nevertheless, it focuses on the general population instead of on the individual. Once cancer, one of the leading causes of global death, is perceived as an isolated incident, we believe macro-environmental circumstances, and not only biological ones, must be considered. Furthermore, cancer in the elderly intensifies solicitude, and expanded policies and actions demand individual health determinants. In the Portuguese Public Health Policy, we started a collaborative Oncology, Human Kinetics, and Public Health Policy project. This is the first project of the Public Health Policy Lab from the Institute of Social and Political Sciences of the University of Lisbon. Based on a brief review of two research projects on improving cancer patients' health, we promote micro-organisational projects to deal with the social phenomena of loneliness, physical activity, and lifestyle. As a sequence of the well-known social determinants, we endorse political determinants as the basis for public health. The latest worldwide governmental trend is to create public labs as an innovation of political policymaking. Throughout this reflection, the need for a new rational approach specially designed for a social model is considered.

**Keywords:** public health; political determinants; social determinants; public lab; cancer; elderly; loneliness

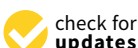

The centralised medicine model (that is, hospital medicine) was developed by Descartes, particularly in his scientific model, according to which, a hospital is an ideal place for treatment [1]. The biomedical model relates the concept of health to the absence of disease [2]. Our health/disease systems focus on treating diseases to promote well-being and health [2]. The biomedical model's biggest problem is that other important and highly influential health state aspects are not considered or well thought out. Public health has evolved into a more holistic concept and now incorporates emotional, social, psychological, and economic factors, and considers how these affect health [3]. The biopsychosocial model involves a more complex approach than that of the biomedical model, suggesting that health and disease concepts cover biological, psychological, and social aspects [1]. WHO emphasises the importance of these and other factors in its definition of health as a state of mental, physical, and social well-being (not just an absence of disease or suffering) [4]. By emphasising the interaction between the environment and the individual, this definition highlights how likely all environmental factors are to affect health [3,4].

Epidemiology is the study of how diseases and their determinants are distributed among the population. Studying the health/disease process using the epidemiological method involves an element of social determination. It implies that societies evolve according to their laws, which go beyond the mere clinical method scope, as they have not been envisaged strictly within the confines of that method's parameters [5]. Dahlgren and Whitehead [6] have developed a model representing all influencing layers to better understand individual health and its determinants. Once the most important determinants are external to the health sector, adopting more broadly scoped policies will ultimately contribute to public health promotion. Nancy Krieger [3] introduces an element of intervention by defining a social determinant as any factor or mechanism that ultimately causes social conditions that affect health (and information-based actions can potentially modify that) [3].

Several initiatives have been launched to ensure that governments everywhere see health as a priority within their internal policies (not only the health-related ones but also those in the economic, social, cultural, and environmental ranges). These initiatives help alert policymakers to a change of mindset, showing the urgent need to improve disease prevention and treatment, and expand policies and actions on promoting health and well-being [7].

Social policy can be understood as a state action in the pursuit of individual well-being. According to Caeiro [8], social policy, as far as public health and hygiene are concerned, includes a set of measures aimed at anticipating and responding to individual and social issues. For the same author, equity and justice are two key concepts when approaching social policy—and these are precisely the concepts that should be substantiating and underpinning that policy [8]. The concept of public health defines organised measures to promote health and well-being, prevent disease, and prolong the general population's lives [4]. Thus, considering that the definition of public health encompasses the health and well-being of the general population—rather than focusing on the individual—this concept is then associated with each government's nation-wide action undertaken [5]. Typically, a policy will take shape as a regulation, a law, an administrative action, a procedure, a voluntary practice or an inducement from a government or any of several other institutions (and political decisions often associated with the allocation of resources will be involved). In the public health context, policy adoption and implementation will include the proposal, launch, and enactment of voluntary practices, regulations, or laws designed to influence public health system development and health promotion [8]. Thus, it is more than obvious how critical each policy adoption can be in this specific sector, especially when considering the final goals: developing public health systems and promoting health [8]. The World Health Organization [9] (WHO) plays a multi-level governance role so that all public health authorities may operate nation-wide based on the same information and with a common goal: developing public health policies [9,10].

The integration of social determinants, such as human relations and solidity in patients with cancer, is a new vision that interconnects health and social factors. Physical exercise in cancer survivors and the analysis of quality of life and longevity in cancer patients are significant features of current oncology. Cancer is one of the leading causes of death in the world. Historically, cancer has been considered an isolated event. However, this framework is limited, since it focuses primarily on cancer biology and less on macro-environmental conditions, which are extremely critical factors in forming this type of disease [5,11]. Access to medical oncology changed, in a positive way, the survival of cancer patients. The health care system should support the patients at all stages of the disease. The patient's social conditions are an important issue in oncology, which impact overall survival and quality of life. Medical oncologists understand the role of political lobbying and innovation and try to adopt this role as an advocate for the patients and oncology [12]. That is why longevity, loneliness, and human relations are central and topical issues in public health policies. In her article *Social Relationships and Physiological Determinants of Longevity Across the Human Life Span*, Claire Yang spots the presence of a

direct correspondence between social and quality-of-life aspects, linking social conditions and their impacts on well-being [13]. By analysing global-level social determinants, Dan Buettner's [14] *The Blue Zones* proves that the highest percentage of individuals over 100 years of age occurs in five zones worldwide. In their article *Loneliness, Health, and Mortality in Old Age*, Ye Luo, Louise Hawkley, Linda Waite, and John Cacioppo conclude that feelings of loneliness increase the risk of mortality [15]. In *The Harvard Gazette*, Liz Mineo [16] reveals the determinants of longevity, notably the most important one: human relations. In fact, Jo Cox's report [17] was at the genesis of Theresa May's government's Ministry of Solitude in the United Kingdom. Steven Cole, John Capitanio, U. C. Davis, and John Cacioppo have also published essential findings on loneliness, greatly influencing the academic and scientific community [18].

The interconnection of the social dimension in health, specifically elderly oncology patients, is a new medical strategy valued by political options [7]. Sequentially, we suggest the concept of "health innovation" as a change to the medical/medicine model. Health innovation can be the valorisation of the social dimension in health. The macro-environment, formed by human relations and projects for solitude, can incorporate physical exercise. These social determinants are targets of political strategy and are considered political. Health innovation considers prioritising political determinants, especially over loneliness, that impact patients' health. Political determinants in the social area, formed by human relations and solidarity, will significantly impact the elderly with cancer. Prioritisation in health is a new form of health innovation. Health policy universities and their laboratories can play a critical role in health policy by changing priorities and becoming new political actors. Innovation generally refers to implementing new products, significant improvements in existing products, new processes, new organisational methods, or new business models [19]. Concerning social innovation in health care, health service users' views play a crucial role in reforming these organisations, especially in terms of economic efficiency. These user opinions help improve the quality standards of services provided [19]. Such improvement occurs through structure-wise and process-wise organisational innovation [19,20]. There are four types of social innovation: social movements, service-related social innovations, social enterprise, and digital social innovations [20]. The social pillar needs a new vision of interconnecting social and health factors for the sustainability of the democratic society [8]. The social dimension in health has become a political priority with an interconnection between health policy and social policy [8]. Elderly patients with cancer need a policy on social determinants to access the best oncology hospital health care [7].

The nudge concept, for instance, stems from the interconnection between economic sciences, political theories, and behavioural sciences [21]. This concept proposes positive reinforcements (and indirect suggestions) on how to influence the behaviour of groups of individuals who hold decision-making authority. Thus, the nudging process (touch process) contrasts with other means of obtaining responses, such as information, education, legislation, or enforcement. This concept was popularised in 2008 by University of Chicago professors Richard Thaler and Cass Sunstein in *Nudge: Improving Decisions About Health, Wealth, and Happiness*. Currently, the nudge theory has been influencing Anglo-Saxon politicians. It debates whether the nudge theory is a new development in behavioural economics or merely new terminology used within the scope of behaviour-influencing methods studied by behavioural analysis sciences [21].

Oncology is an area of medical research where researchers continually focus on finding genetic information that explains and/or solves each oncological nature problem [11]. Oncology also works in prevention, diagnosis, primary health care, and health promotion [11]. Health promotion is critical to the quality of life of cancer patients [11]. In cancer patients, the social component plays a vital role in choosing diagnostic strategies, prevention, and treatment [11]. The importance of this aspect is felt mainly by the elderly [11,22]. Sport, as a generic area or domain, includes several physical activities, and the attitudes towards this social phenomenon are multiple. Sport is an integral part of our social life and imperative within the political, cultural, and economic systems. As Sarmento [23] has said, sport is

a form of human behaviour that can be explained by external and internal signals and the laws of a society's organic systems. Sport is also associated with the scientific view of movement.

Sport is a social determinant that can be supported by social policy options [20]. Overall, health policies should consider sport as a prevention and treatment tool for cancer patients. In a human mobility logic, specific exercises for cancer patients should be considered and studied, bearing in mind that this procedure must be customised according to the different types of cancer and the specificity of the medical treatment to which the patient is subjected [24]. This strategy requires specific types of exercise and individualised prescriptions. Generally, patients with cancer pathology should have an exercise's prescription. A proper assessment and a choice of a good option will provide the basis for prescribing physical activity in cancer patients [25]. Assessing the economic impact of these options is also necessary [24]. Social innovation used to adopt exercise and loneliness strategies can be necessary for the cancer patient (especially for elderly patients, as many studies have shown). To reiterate, innovation in projects based on the nudge theory (i.e., behavioural modulation projects in public health policy) can be the basis for pilot projects in the field of cancer-targeted social health policy [21].

Furthermore, we briefly discuss two research projects to explain how physical activity or exercise can affect cancer patients' health, particularly regarding the multiple determinants involved (the highlight going to loneliness). These projects (or interventions) stand on the premise that health systems must treat several diseases and promote healthy habits among the population to ensure their well-being, health, and quality of life [26]. Both projects are concerned with cancer patients (in the first case) and cancer in the elderly (in the second case), emphasising the issue's social side.

The first project, the clinical study "Evaluating and Following Breast Cancer Patients in a Better Way", presents a new kind of breast cancer assessment. This new form of evaluation is innovative, particularly in addressing variables such as bone health, sleep patterns, and heart function, and not just variables related to disease biology. Greater attention is paid to patient assessment rather than to the tumour's biological characteristics [27].

The second project, still being handled at the Lab of Public Health Policy of the University of Lisbon (ISCSP-UL), examines the extent to which patients can share responsibility for their illness by assessing their habits, behaviours, and social relations. It proposes a behavioural change in the sociability model by introducing the "ball exercise" in elderly patients. One can then try observing how this change might relate to the quality of life of the elderly. In cancer patients, particularly the elderly, there is evidence that sports, human relations, and the social component justify the political agenda. There are behavioural pilot projects, such as the so-called "cancer health policy", that can positively impact these patients' longevity and quality of life. They help make both health systems and society more resilient in the face of cancer and turn specific behavioural pilot projects into primary experimental public policy tools.

According to the Portuguese author and scholar Isabel Fonseca [28], culture organises and adjusts all policies and types of governance adopted. The projects submitted are cultural, relate to lifestyles, and are based on rational grounds to combine society's knowledge, health sciences, public health and social policies, and human kinetics. General science designs a health-oriented culture [1]. The underlying policy is a macro-organisational one and results from the scientific knowledge provided by quantitative and qualitative studies. Undoubtedly, culture can be valued by social innovation strategies [20]. For some public policy thinkers, such as Laswell [11], the experimental method has always played an important role. The submitted micro-organisational projects dealing with health-related political and social phenomena (loneliness, physical activity, and lifestyle) can be valued whenever public policies are implemented, either because of their rationale or behavioural studies [29]. It takes proper and accurate foresight to see beyond the social determinants of health. Indeed, these are the most fundamental drivers of population health and health equity. Nevertheless, health's political determinants are far more impactful, and the social

determinants' real impulse. Wise public policies and support are central pillars of the turn society needs [30]. By shaping policies, politicians reshape communities' equity health despite critical strengths. So, it is crucial to recognise, address, and combat health inequity by endorsing a strategic health model. Through inter-collaboration between political decision-makers, we can create new tools to reduce health inequity in every community.

Despite the awareness of the Public Sector Innovation (PSI) Labs by policy scientists, governments are seeking new procedures for public policy and service design [31]. These original and unique "change agents" of the public sector are developing shortcomings to apply to conventional programs. It is relevant to comprehend whether its new ideas, proposals and methods conceive an innovative difference to the public sector's reconfiguration. Usually, these labs are independent of the rest of the public sector and operate their purposes and methods autonomously, although they are sometimes quite near the executive power. They do not perform time-limited projects but target the ideal of solving social and political obstacles [31]. They base their solutions upon social scientific evidence rather than ideology or beliefs. While some PSI Labs are non-governmental agencies engaged in public policy and public sector innovation, others stand either within the executive government or operate in governmental agencies. The one characteristic common to all is public policies' design through innovative methods that correlate all stakeholders' processes. We believe that citizens' health equity awareness and the formulation of solutions without politic persuasion might improve public service innovation policies.

In sum, the biomedical model has been the basis for the evolution of medical science and the focus on disease is the current paradigm in medical schools. The biosocial model and health promotion are the new future in medicine. Public health is responsible for health policies and health promotional strategies, while solitude and longevity are the new public health policies. Longevity is causally related to human relations and their relationship with the macro-environment. Understanding longevity as the quality of life and survival is essential for elderly patients with cancer. Health policies for longevity and loneliness are also current issues within the health policy. We approach health policies for longevity in cancer patients with behavioural clinical trials that allow their adaptation to loneliness during the macro-environment factors' effects. Loneliness in geriatric cancer patients is significant and must be thoroughly discussed by health policies.

Resolutions shall address all the determinants of health to improve the so-far excluded people's quality of life. Moreover, scientific research shall discern if the PSI Labs are making a positive difference to public health policies or not. Nevertheless, by reframing health circumstances by looking at health's political determinants, we will get healthier communities, because change matters.

**Author Contributions:** All authors contributed to the design and implementation of the study, to the analysis of the results and critical revision. V.F. collected and interpreted the data, conceived the research, and drafted the manuscript.; J.C. and F.N. critically revised the final manuscript. All authors have read and agreed to the published version of the manuscript.

**Funding:** ISCSP—Instituto Superior de Ciências Sociais e Políticas da Universidade de Lisboa–CAPP—Centro de Administração e Políticas Públicas. This work was supported by Portuguese national funds through FCT - Fundação para a Ciência e a Tecnologia, under project UIDP/00713/2021.

**Institutional Review Board Statement:** Not applicable.

**Informed Consent Statement:** Not applicable.

**Data Availability Statement:** Not applicable.

**Conflicts of Interest:** The authors declare no conflict of interest.

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
