# Peer review of "Social Model—Innovation and Behavioural Intervention as a Public Policy of Action within an Oncology and Loneliness Scope"

_sustainability, doi:10.3390/su13031544_

Round 1

Reviewer 1 Report

The proposed analysis concerns the investigation of innovation and behavioural intervention in medical field.

It is really interesting but some points have to be clarified. In particular, it concerns the analysis of social model by using AI/Deep Model (see An Epidemiological Neural network exploiting Dynamic Graph Structured Data applied to the COVID-19 outbreak. IEEE Transactions on Big Data) and diffusion models (see Diffusion algorithms in multimedia social networks: A preliminary model. In IEEE/ACM International Conference, 2017; pp.844-851).

Finally, I suggest a linguistic revision.

Author Response

Dear Reviewer, 

Firstly we would like to thank you for your comments and suggestions, which helped us to improve this review. We believe we have clarified the main points of this article and also made a linguistic revision. 

We have consulted the links suggested for the social model (Al/Deep Model and Diffusion Models) and we find it quite interesting for succeeding research.  

Kind regards,

Vasco Fonseca

Reviewer 2 Report

The paper does not fulfil the criterion of a scientific text. It is neither an original paper nor the review (the type not indicated at the top of the first page). The abstract is lacking, as well as the objective of this study. Rather, it can be regarded as an opinion expressed by the authors. A significant part of the article is made up of successive definitions of various concepts relating to public health. The main idea of this paper is difficult to catch among these definitions. The reference to innovation suggests an unconventional approach to intervention programs and hospital (?) practice in oncology. Meanwhile, the authors attempt to present as their own discovery the biopsychosocial model of health well known for about 40 years.  Moreover, it is also difficult to distinguish which comments are global and which relate to Portuguese realities.  The concept of work should be put in order with a clear objective. It would be worthwhile to place this work in the evolution of the approach to understanding the concept of health, with an emphasis on what is really new and innovative. 

Author Response

Dear Reviewer, 

Firstly, we would like to thank you for your suggestions that helped us improve this document. 

On the first submission, both the type of paper and abstract were missing. That is now corrected, plus a revision to the article to develop the objective, and to clarify both the social model's approach and the innovation's suggestions. We have clarified that Portugal pictures Europe, and follows European trends. As suggested, we have changed the order of some parts of the text to help understand its object. The conclusion shows what are the latest tendencies in this topic.  

Kind regards,

Vasco Fonseca

Round 2

Reviewer 1 Report

I think that the authors have addressed all my concerns.

Author Response

Dear Reviewer, 

We would like to express to you our thankfulness for your kind suggestions that we have tried to answer during the two revisions. 

Kind regards,

Vasco Fonseca

Reviewer 2 Report

The main message coming from this paper is more transparent in its present form. However, the presentation is still not comprehensible. It resembles an essay or an editorial. It lacks any structure. The authors jumped between topics from the problems of elderly and disabled people to physical activity and illustrated the theoretical considerations with an example on breast cancer.

Neither in the introduction nor in the abstract the purpose of this paper is stated. Difficult to determine at all time whether a new authorial concept is included in this text. The paper needs a standard distinction between what is already known and what the authors propose new. A very interesting reference is [19], which defines social innovation by giving its 4 components. Such a definition is missing in this text. Instead of, the term health innovation appears. Is this healthcare innovation or an extension of the definition of health?

You are kindly asked to look at this paper from the side of a reader who knows little and is trying to learn something new and understand the essence of this approach. It is necessary to decide whether the work is addressed to people who know the subject very well (as an expert debate) or to people who are less familiar with this topic.  

Author Response

Dear Reviewer,

We would like to thank you for your kind suggestions towards the revision, which we have tried to accomplish this time. 

Concretely, to clear this study and the sequence of topics, we have changed the abstract on the first revision. So, we expect the presentation to be more comprehensive than before. The biomedical model introduces the approach to this article's main problem: finding out an innovative way to deal with the social consequences of elderly patients with a specific disease, cancer. The main targets are to reduce loneliness while improving human relations and longevity. Consequently, we mention social policies and determinants, plus its interconnection to Health. Then, addressing the need for social innovation in healthcare is inducing a reform of the intervening organizations. It takes some information concerning oncologic patients to lead the readers to the influence of habits' change and the practice of exercise during treatment. We also present two research projects. The first one proves the significance of external variables to the success of the disease's recovery. The second project is a work in progress towards a quality of life improvement. We have also mentioned the four social innovation pillars, as suggested.
We believe the paper is now clear both to experts or ordinary readers as recommended. 

Kind regards,

Vasco Fonseca